# Preparation of a Montmorillonite-Modified Chitosan Film-Loaded Palladium Heterogeneous Catalyst and its Application in the Preparation of Biphenyl Compounds

**DOI:** 10.3390/molecules27248984

**Published:** 2022-12-16

**Authors:** Zhifei Meng, Zijian Wang, Yu Li, Wei Li, Kewang Zheng, Zufeng Xiao, Wei Wang, Qin Caiqin

**Affiliations:** 1School of Chemistry and Materials Science, Hubei Engineering University, Xiaogan 432000, China; 2Hubei Biomass–Resource Chemistry and Environmental Biotechnology Key Laboratory, School of Resource and Environmental Sciences, Wuhan University, Wuhan 430000, China

**Keywords:** polyvinyl alcohol, montmorillonite, chitosan film, heterogeneous catalytic materials, biphenyl compounds

## Abstract

The natural polymer chitosan was modified with polyvinyl alcohol to enhance the mechanical properties of the membrane, and then, the montmorillonite-modified chitosan-loaded palladium catalyst was prepared using the excellent coordination properties of montmorillonite. The results showed that the catalyst has good tensile strength, thermal stability, catalytic activity, and recycling performance and is a green catalytic material with industrial application potential.

## 1. Introduction

Biphenyl derivatives can be widely used in the preparation of organic synthesis intermediates [1,2,3], chiral skeletons of asymmetric catalysts [4,5], organic functional materials [6,7], liquid crystals [8,9], etc. The Suzuki coupling reaction [10,11,12] has the advantages of the economical and easy availability of boronic acid derivatives as substrates, mild reaction conditions, and tolerance to a variety of active functional groups, and it is one of the most important methods for the synthesis of biphenyl derivatives.

Compared with transition metals such as Au [13,14,15], Ag [16,17,18], Pt [19,20,21], Ru[22,23], Rh [24,25,26], and Cu [27], Pd [28,29,30] is the catalyst of choice for the Suzuki coupling reaction due to its many advantages, such as high catalytic efficiency, high selectivity, and low dosage. However, most of the palladium catalysts are mainly homogeneous catalysts, which are easily left in the product and contaminate the target product, and these homogeneous catalysts are poorly recyclable, easily lost, and easily deactivated. The preparation of efficient, stable, and environmentally friendly heterogeneous catalysts by immobilizing transition metals on some inert materials to overcome the disadvantages of homogeneous transition metal catalysts, due to difficult separation and recycling and their pollution of the environment, is gaining more and more attention.

Both chitosan (CS) and montmorillonite (MMT) are abundant natural materials in nature and are typical representatives of polymeric and inorganic carriers for transition metal catalysts. Chitosan [31,32,33,34] is a product of chitin deacetylation and has advantages as a polymeric carrier: it is insoluble in common solvents and suitable for catalyst separation and recovery. Due to a large number of hydroxyl and amino groups on the chitosan molecular chain, it is easy to be chemically modified and has a strong coordination ability with transition metal ions. Montmorillonite [35,36,37] also known as kaolinite, is a natural silicate material, which is a flake formed through the accumulation of nanoscale silicate flakes with a negative surface charge through electrostatic action. Montmorillonite has excellent adsorption and thermal stability properties, and when used as an additive to nanopolymers, it can substantially improve the mechanical and thermal stability properties of polymers. The chitosan intercalation-modified montmorillonite composites retain the good complexation ability of chitosan polymers with transition metals, but also combine the large specific surface area, high thermal stability, and strong molecular adsorption properties of the montmorillonite matrix, and these are ideal carrier materials for the preparation of heterogeneous catalysts.

In previous work, we immobilized metallic copper on chitosan microspheres and found that it has high catalytic activity in the self-coupling reaction of phenylboronic acid [38]. However, the recovery of chitosan microspheres was more troublesome; however, the incorporation of chitosan into membrane materials can effectively solve the recycling challenge. The addition of polyvinyl alcohol (PVA) to the catalytic carrier material can effectively improve the mechanical properties of the material by taking advantage of its excellent film-forming properties. Here, CS/PVA/MMT@Pd catalytic material was prepared, and its catalytic activity was investigated for the Suzuki coupling reaction in the aqueous phase. Then, the structure of the membrane materials was examined via infrared spectroscopy, transmission electron microscopy, and thermogravimetric analysis, and the catalytic properties, substrate suitability, and reusability of the catalysts were studied with the Suzuki reaction. The results showed that Pd was highly dispersed in the membrane material, with good substrate tolerance and catalytic activity, and the catalyst was easily separated and reused at least six times.

## 2. Results and Discussion

### 2.1. Mechanical Properties of CS, PVA, CS/PVA, and CS/PVA/MMT Film

The CS, PVA, CS/PVA, and CS/PVA/MMT film materials were tested with a Universal tensile tester, the results of six tests were averaged, and the values are listed in Appendix A. Figure 1 shows the mechanical properties of CS, PVA, CS/PVA, and CS/PVA with different MMT mass ratio films. The addition of PVA to chitosan membranes can effectively improve the mechanical properties of the membranes [39]. We used a CS:PVA = 1:1 film as the reference film and added montmorillonite gradually to the reference film; the tensile strength and elongation at break showed a decreasing trend with the addition of MMT. When the ratio of montmorillonite was increased to 1:1:0.5, the mechanical properties of the CS/PVA/MMT film were close to those of CS, which might be because the addition of montmorillonite destroyed the crystal structure of PVA during the mixing process [40,41]. Considering the mechanical properties and adsorption properties of the films, the CS/PVA/MMT film with a mass ratio of 1:1:0.2 was selected as the carrier of the catalyst.

### 2.2. Dynamic Adsorption of the CS/PVA/MMT Film

The adsorption kinetics reflect the rate of the adsorption reaction and the relationship between the adsorption amount and time. The common adsorption kinetic fitting models are the pseudo-first-order model, pseudo-second-order model, and Weber Morris model. The formula expressions are shown in (1), (2), and (3), respectively, where *q_e_* = equilibrium adsorption capacity, *q_t_* = adsorption capacity, *k* = rate constant, *t =* time, and *c_i_* = concentration of Pd^2+^ at time *t_i._*
(1)$$\mathrm{lg}\left(q_{e}-q_{t}\right)=\mathrm{lg}q_{e}-\frac{k_{1}}{2.303}t$$
(2)$$\frac{t}{q_{t}}=\frac{1}{k_{2}q_{e}^{2}}+\frac{t}{q_{e}}$$
(3)$$q_{t}=k_{2}t^{2}+c_{i}$$

When the pH was above 3, Pd^2+^ was rapidly hydrolyzed and even resulted in PdO precipitation [42]. Therefore, the adsorption at pH ≤ 3 was investigated in this experiment, and the results of Pd^2+^ adsorption by the CS/PVA/MMT membrane are shown in Table 1. First, the adsorption performance of the membranes was studied with the pH of the solution (Table 1, entries 1–3); setting the time of adsorption to 300 min at pH = 3, the adsorption of the membranes was 94.91 mg/g, while with a pH = 1, the adsorption of the membranes decreased rapidly to 39.91 mg/g. The variation in Pd^2+^ adsorption may be related to a large number of amino groups, as well as hydroxyl groups, present on the adsorbent. At pH ≤ 3, Pd^2+^ exists mainly as [PdC1_4_]^2−^, most of the amino groups in the adsorbent are protonated, [PdC1_4_]^2−^ and protonated amino groups undergo electrostatic adsorption, and the anions (Cl^−^) in solution compete with [PdC1_4_]^2−^ for the positively charged protonated amino adsorption sites [43]. Therefore, the lower the pH, the stronger the competition by Cl^−^, and the lower the adsorption of Pd^2+^. The adsorption capacity of the membranes in solutions with different concentration gradients was studied by setting the pH = 3; 20 mL solution was used at room temperature, the mass of the membrane was 20 mg, and the adsorption time was 300 min (entries 4–7). The adsorption capacity of 20 mg chitosan microspheres for Pd^2+^ reached a maximum of 94.91 mg/g at an initial Pd^2+^ concentration of 100 mg/L. The adsorption capacity of the membranes increased with time as shown in entries 8–16. The adsorption was nearly completed at 240 min and reached adsorption equilibrium at 300 min. Therefore, the optimum adsorption conditions at room temperature are: solution pH = 3, adsorption time = 300 min. Compared with the CS/PVA film (entry 17), the CS/PVA/MMT film showed a better adsorption effect, indicating that the addition of montmorillonite increased the adsorption performance of the membrane.

The pseudo-first-order reaction model, pseudo-second-order reaction model, and Weber Morris model were used to perform a linear fit with the experimental data in Table 1. As shown in Figure 2, Appendix A, the experimental data were more consistent with the fitted results of the pseudo-second-order reaction model. According to the fitted model, the maximum adsorption amount of chitosan microspheres can reach 104.06 mg/g, which is close to the experimentally measured equilibrium adsorption amount. The fitting results indicate that the internal diffusion of Pd^2+^ is the main reason affecting the adsorption [44].

### 2.3. Optimization of Reaction Conditions

Liu et al. [45] have found that the Suzuki coupling reaction in air or oxygen is much faster than that in nitrogen, and thus, this experiment was not protected by the addition of nitrogen. The effect of the solvent type, base type, reaction temperature, and reaction time on the reaction was investigated using CS/PVA/MMT@Pd as the catalyst and 3-methoxybromobenzene (**1a**) and 4-methoxybenzeneboronic acid (**2a**) as the template substrates. The results of condition screening are shown in Table 2. The effect of the solvent type on the reaction was first screened (entries 1–7). Among the protic solvents, higher yields of **3a** were obtained with MeOH (59%), EtOH (55%), and i-PrOH (50%). In contrast, among the aprotic solvents, the yields were lower than 22% with other solvents, such as MeCN, CH_3_COCH_3_, and CHCl_3_, except for toluene as a solvent, which resulted in a higher yield (42%).

When pure water was used as the reaction solvent (entry 8), the feedstock did not react. The mixed solution containing water increased the solubility of the inorganic base and promoted the reaction [46,47]. Then, different ratios of organic solvents to water were screened as seen in entries 9–16. When the ratio of EtOH to water was 2:1 (*v*/*v*), the yield of the product could reach 92%. The carbonates (M_2_CO_3_, M = Li, Na, K, and Cs) were all effective in promoting the reaction (entries 17–20), and yields of 95% could be achieved when CS_2_CO_3_ was chosen as the base for the reaction. When the amount of base was increased to 2 times the equivalent, the yield of **3a** did not increase significantly, and thus, we chose 1.2 times the amount of base (entries 21–22). Finally, the effects of the reaction temperature, reaction time, and catalyst dosage on the reaction were investigated (entries 23–32). Finally, the optimal conditions for the reaction were obtained as follows: 3-methoxybromobenzene (**1a**, 0.3 mmol), 4-methoxyphenylboronic acid (**2a**, 0.36 mmol), CS/PVA/MMT@Pd (10 mg, 3mol%), Cs_2_CO_3_ (0.36 mmol), 70 °C, and 10 h.

The applicability of CS/PVA/MMT@Pd catalysts in the Suzuki reaction was investigated under optimal reaction conditions as shown in Table 3. The reactivity of different bromine substitutes was studied first using 4-methoxyphenylboronic acid as the boronic acid reagent. When o-, m-, and p-bromotoluene were used as substrates, respectively, **3b-d** was obtained in higher yields (90–95%). Lower yields were obtained when 2,6-dimethyl-1-bromotoluene was used as the reaction substrate compared to those with o-bromotoluene, probably due to the Steric effects. **3f**–**3k** was also obtained in relatively high yields (68–92%) when brominated benzenes with absorbing electrons, such as cyano, formyl, ester, methyl hydroxyl, or amino groups, were used. **3m**–**3p** was also obtained in high yields (52–92%) when heterocyclic halogenated hydrocarbons, such as 4-methyl-2-bromopyrimidine, 3-romoquinoline, and 3-bromothiophene, were used as reaction substrates. Subsequently, **3q**–**3z** was obtained in relatively high yields (67–96%) when phenylboronic acid, 4-cyanophenylboronic acid, and 4-tert-butyl phenylboronic acid were selected as boronic acid reagents. These experimental results indicate that the CS/PVA/MMT@Pd catalysts have good adaptability to different substrates. The ^1^H NMR and ^13^C NMR spectra of the products (**3a**–**3ab**) are shown in Appendix A.

The cross-coupling of aryl chlorides is difficult because aryl chlorides show very low reactivity. Under the optimized conditions, the chlorine substituents did not react when 1-chloro-4-bromobenzene (1h) and 4-bromo-2-chloroaniline (**1k**) were used as reactants. By increasing the amount of phenylboronic acid(2eq), switching to DMF as the reaction solvent, and increasing the temperature of the reaction, the coupling product of aryl chloride was obtained. The yields of the obtained products are shown in Table 4.

### 2.4. Gram-Scale Synthesis Reaction

The application of the CS/PVA/MMT@Pd catalytic material to the gram-scale reaction allowed us to investigate the feasibility of this material in large-scale applications. The reaction of 3-methoxybromobenzene (10 mmol) and 4-methoxyphenylboronic acid (12 mmol) under the optimized conditions yielded 1.99 g of biphenyl product at a 93% yield, which indicates that the material has some potential for large-scale applications.

### 2.5. Catalyst Reuse

The recovery of the catalyst is an important indicator for evaluating heterogeneous catalytic materials. The recovery experiments were carried out on CS/PVA/MMT/Pd membrane materials under optimized conditions. The used catalysts were washed three times with water and ethanol alternately under ultrasonic conditions and then directly entered the cycling experiment. The catalytic activity of the catalyst was slightly decreased after six cycles of the cyclic reaction (Figure 3), which indicated that the catalytic material has good reusability.

### 2.6. Characterization of Catalytic Materials

The microstructures of the catalytic materials were analyzed using infrared spectroscopy (FIIR), thermogravimetric analyzer (TG), polarization microscopy (PM), X-ray diffraction (XRD), transmission electron microscopy (TEM), and X-ray photoelectron spectroscopy (XPS) test instruments.

#### 2.6.1. FTIR Characterization

Infrared spectroscopy can be used to confirm the structure of catalyst film materials and the interaction forces between groups within the materials. The infrared spectra of CS films are shown in Figure 4a. The peaks formed by the stretching vibrations of υ(-OH) and υ(-NH_2_) were at 3350 cm^−1^, the absorption peaks of the stretching vibrations of υ(-CH_2_) were at 2926 cm^−1^, the absorption peaks of the amide II band (δ(C-N) and δ(N-H)) deformation vibration absorption peak and amide III bands (υ(C-N) and δ(N-H)) deformation vibration absorption peaks had characteristic absorption peaks at 1614 cm^−1^ and 1306 cm^−1^, respectively, and the υ(C-OH) stretching vibration absorption peak was at 1036 cm^−1^. In the IR spectrum of CS/PVA films, as shown in Figure 4b, the υ(-OH) and υ(-NH_2_) stretching vibration peaks in CS/PVA films shifted to lower wave numbers compared to those of CS films. The amide I band of chitosan (υ(C=O)) appearance and the υ(C-OH) stretching vibrational absorption shifted from 1036 cm^−1^ to 1089 cm^−1^ due to the addition of PVA, indicating a decrease in the proportion of amino groups in the film. The disappearance of the crystalline sensitive peak of chitosan at 668 cm^−1^ indicated that the addition of PVC disrupted the crystalline structure of chitosan. As shown in Figure 4c, the IR spectra of CS, PVA/PVA/MMT, and CS/PVA co-blended films were very similar, with a υ(Si-O-Si) stretching vibration peak at 1029 cm^−1^, which is a characteristic peak of montmorillonite. As can be seen in Figure 4d, the intensity of the IR spectral peaks of the solidly loaded Pd co-blended film diminished compared to that of the un-solidly loaded Pd film, but the position of the peaks remained essentially unchanged.

#### 2.6.2. Thermogravimetric Analyses

The thermal stability of the membranes and catalysts was analyzed with test conditions set to a heating rate of 10 °C/min at N_2_ flow (20 mL/min). The thermal decomposition diagrams of CS, CS/PVA, CS/PVA/MMT, and CS/PVA/MMT@Pd films are shown in Figure 5. The weight loss at temperatures below 130 °C is the evaporation of small molecules of water remaining in the films. The maximum weight loss of CS films was at 150 °C, which corresponds to the thermal decomposition temperature of the chitosan backbone. The maximum weight loss temperature increased to 200 °C for the CS/PVA and CS/PVA/MMT hybrid films, indicating that the films became more thermally stable. The thermal loss diagram of CS/PVA/MMT@Pd films was similar to that of CS/PVA/MMT membranes, indicating that the catalyst with solid palladium has good thermal stability.

#### 2.6.3. Polarization Microscopy Analyses

The membrane material was placed on a slide, and the light brightness was gradually adjusted from weak to strong, with the measurement conditions being 200× magnification, in reflection mode. From Figure 6, it can be seen that the palladium metal particles were uniformly dispersed in the membrane before the reaction, and the palladium particles were also more uniformly dispersed in the membrane after one reaction, except for a few particles that were aggregated.

#### 2.6.4. XRD Analysis

It can be seen from Figure 7 that CS/PVA had a strong diffraction absorption peak at 2*θ* = 17.0° and weaker diffraction absorption peaks at 2*θ* = 19.0° and 28.8°, respectively. The intensity of the diffraction peak in CS/PVA/MMT decreased, indicating that the addition of MMT leads to a decrease in the crystallinity of polyvinyl alcohol/chitosan crystals. The XRD patterns of the membranes did not change significantly after the addition of palladium, indicating that Pd was highly dispersed in the membranes. As shown in Appendix A (Appendix A), the diffraction angles of the reacted membrane material membranes did not change, indicating that Pd did not aggregate on the membrane surface.

#### 2.6.5. XPS Analysis

The changes in the chemical valence of Pd before and after the reaction can be described by the XPS spectra. As shown in Figure 8a, two characteristic peaks of Pd^0^ appeared in the newly prepared CS/PVA/MMT@Pd with electron binding energies of 340.18 eV (3d_3/2_) and 334.98 eV (3d_5/2_), indicating that the catalytic material containing Pd^0^ was successfully prepared. After six reaction cycles of CS/PVA/MMT@Pd (Figure 8b), in addition to the still present characteristic peaks of Pd^0^ with electron binding energies of 340.98 eV (3d_3/2_) and 335.98 eV (3d_5/2_), respectively, characteristic peaks of Pd^2+^ also appeared with electron binding energies f of 342.58 eV (3d_3/2_) and 337.08 eV (3d_5/2_), indicating that the oxygen present in the reaction oxidizes the pd^0^.

#### 2.6.6. TEM Analysis

Figure 9 shows the TEM images of the unreacted CS/PVA/MMT@Pd film and the CS/PVA/MMT@Pd film after one reaction cycle. It can be seen from Figure 6a,b that the size and dispersion of the metallic palladium particles on the surface of the chitosan film were relatively uniform, which is consistent with the test results of polarized light microscopy. The lattice spacing of metallic Pd particles was about 0.23 nm, as tested via HR-TEM, which corresponds to the (1,1,1) crystallographic plane of the face-centered cubic-structure Pd. These results indicate that the catalytic material has good reusability properties.

### 2.7. Discussion

As shown in Table 5, many catalysts were applied to the Suzuki coupling reaction with good experimental results. Moniriyan et al. [48] have obtained biphenyl products with a 74% yield using Pt-APA@Fe_3_O_4_/GO as a heterogeneous catalyst with DMF as the reaction solvent. Kumar et al. [49] have obtained biphenyl products with a 69% yield using copper complexes at room temperature. Pd is more widely used as an efficient heterogeneous catalyst that can be applied to the Suzuki reaction. In addition to the direct use of Pd or Pd salts as catalysts, such as with PdCl_2_ [50] and Pd NPs [30], Pd or Pd salts is also solidly loaded on materials, such as graphene [51], PICB-NHC [52], polyimide [53], chitosan [54], and starch [55]. The catalytic activity of all of these catalysts was high (89–99%). Compared with the above catalysts, CS/PVA/MMT@Pd is easier to separate and does not require additional ligands, making it a green catalyst suitable for a large number of reactions.

## 3. Materials and Methods

### 3.1. Materials

Chitosan (viscosity, 200–400 mPa.s, deacetylation > 95%), polyvinyl alcohol (M.W. 8200, alcohol solubility: 85.0–90.0 mol%), Na-montmorillonite (>99%), glycerin(AR), acetic acid(AR), NaBH_4_(CP), PdCl_2_(AR), HCl(CP), NaOH(AR), HNO_3_(CP), MeOH(AR), EtOH(AR), i-PrOH(AR), toluene(AR), CH_3_COCH_3_ (AR), CH_3_CN(AR), CHCl_3_(AR), DMF(AR), Li_2_CO_3_(AR), Na_2_CO_3_ (AR), K_2_CO_3_ (AR), Cs_2_CO_3_ (AR), brominated aromatics(**1**, AR), chlorinated aromatics(**4**, AR), 4-methoxyphenylboronic acid(**2a**, AR), benzeneboronic acid(**2b**, AR), 4-tert-butylbenzeneboronic acid(**2c**, AR), and 4-cyanophenyl boronic acid(**2d**, AR) were used. All of the above reagents were purchased from Sarn Chemical Technology Ltd.

### 3.2. Analytical Tests and Characterization

Inductively coupled plasma-optical emission spectroscopy (ICP-OES, IRIS Intrepid II, Thermo, Massachusetts, USA) was used to test the concentration of Pd^2+^. An electronic universal testing machine (AG-IC 5KN, 100KN) was used to test the mechanical properties of the films. Infrared spectroscopy (FTIR, Gangdong 850, Tianjin, China) was used to identify the groups or structures present in the membrane material. A thermogravimetric analyzer (TG, Netzsch, STA449F5, Bavaria, Germany) was used to analyze the thermal stability of the material with test conditions set to an N_2_ flow rate (20 mL/min) and a heating rate of 10 °C/min. Polarized optical microscopy (POM, Zeiss, Axio Scope.A1, Oberkochen, Germany) was used in reflection mode to test the distribution of palladium in the film. X-ray diffractometry (XRD, Bruker, D8 Advance, Saarbrücken, Germany) and transmission electron microscopy (TEM, Japan, JEOL2100F, Tokyo, Japan) were used to analyze the microscopic morphology of palladium in membranes. X-ray photoelectron spectroscopy (XPS, ESCALAB 250xi, Thermo, Massachusetts, USA) was used to analyze the change in the chemical valence of palladium in catalytic materials before and after the reaction. The products were purified via column chromatography (silica gel, 200–300 mesh, Qingdao, China), and their structures were examined based on nuclear magnetic resonance (NMR, Bruker, Avance III 400 Hz, Saarbrücken, Germany).

### 3.3. Preparation of Catalytic Materials

The catalytic material was prepared according to the method of Ref. [57]. Here, 10 mL of 2% CS solution and 10 mL of 2% PVA solution were stirred thoroughly at 60 °C; then, 0.10 g of Na-MMT was added, and stirring was continued for 8 h. The defoamed liquid was poured into a mold and dried to produce CS/PVA/MMT films. Next, 0.0200 g of CS/PVA/MMT film was weighed accurately, and it was added to 20 mL of 100 mg/L palladium chloride solution and shaken for 4 h.

The membrane with sufficient absorption of PdCl_2_ was selected as the catalytic material, the Pd^2+^ in the remaining solution after adsorption was tested by ICP-OES, and the Pd content in the catalyst was 94.89 mg/g. Then, an excess of NaBH_4_ was added and reduced to get a black film. The films were washed with water and ethanol under ultrasonic conditions to neutralize them and dried under a vacuum to obtain CS/PVA/MMT/@Pd catalytic material.

### 3.4. Adsorption Kinetic Experiments with Pd^2+^ (25 °C)

A standard Pd^2+^ solution of 100 mg/L was firstly configured and then diluted to 20 mg/L, 40 mg/L, 60 mg/L, 80 mg/L, and 100 mg/L solutions. The pH of the solution (20 mL) was adjusted using NaOH and HCl, and then, 20 mg of CA/PVA/MMT film was added and shaken at 200 r/min for 5 h. After the adsorption reaction was completed, the clear solution was diluted with 1% HNO_3_ solution at appropriate multiples, and the remaining adsorbent concentration was measured by ICP-OES.

Equation (1) and Equation (2) represent the adsorption percentage and adsorption capacity of the membrane, respectively, where *c_0_
*represents the initial concentration of Pd^2+^ (mg/L), *c_i_* represents the immediate concentration of Pd^2+^ at time *t_i_* (mg/L), m represents the mass of the membrane (g), and *v* represents the volume of the solution (L).
(4)$$Adsorption\%=\frac{\left(c_{0}-c_{i}\right)}{c_{0}}\times 100\%$$
(5)$$q_{i}=\frac{(c_{0}-c_{i})\cdot v}{m}$$

### 3.5. Testing of Mechanical Properties of Film

The test analysis was carried out according to the standard GB1037-88. The flat film was taken and cut into 2.50 cm × 0.4 cm samples. The thickness of each specimen was measured at different places, and the average value was taken as the thickness of the film (*d*). The stretching rate was set at 20 mm/min, each group of samples was measured six times, and the average value was taken to calculate the tensile strength (*δ*, MPa) and elongation at break of the specimen (*ε*, %) according to the following formula.
(6)$$\delta =\frac{F}{l\times d}$$
(7)$$\varepsilon =\frac{l_{1}-l_{0}}{l_{0}}\times 100\%$$
where: *F*—maximum tensile stress, N.

*l*—length of the middle part of the sample, mm.

*d*—average thickness of the sample, mm.

*l_0_*—distance between the original markers of the specimen, mm.

*l_1_*—distance between the markers at the time of fracture of the specimen, mm.

### 3.6. Suzuki Coupling Reaction

3-Methoxybromobenzene (**1a**, 0.3 mmol), 4-methoxyphenylboronic acid (**2a**, 0.36 mmol), Cs_2_CO_3_ (0.36 mmol), and CS/PVA/MMT@Pd (10 mg, 3mol%) were weighed into a reaction tube in air, and 2 mL of mixed solvents was added as the reaction solvent at 70 °C (Figure 10). After the reaction had stopped, the extract was split and spun dry to collect the primary product; then, a mixture of ethyl acetate and petroleum ether (*v*:*v* = 1:100–1:8) was used as a drenching agent, the product was collected and separated via chromatography on a column, and the final product structure was verified through NMR.

## 4. Conclusions

In conclusion, we synthesized CS/PVA/MMT@Pd catalytic materials with Pd uniformly dispersed in the membrane. These materials exhibit high catalytic activity in Suzuki reactions under suitably mild conditions and are suitable for a wide range of aryl bromide substrates. The CS/PVA/MMT@Pd catalytic materials can be separated from the reaction solution through simple manipulation and recovered for recycling. After six cycles, more than 90% yield was still maintained and the material exhibited a strong ability to immobilize palladium. However, the catalytic performance of this material for aryl chlorides was unsatisfactory. In the future, polymers, such as polyurea, lignin, and other polymers, will be investigated as Pd nanocatalyst carriers to further improve the performance of the heterogeneous system.

## Figures and Tables

**Figure 1 molecules-27-08984-f001:**
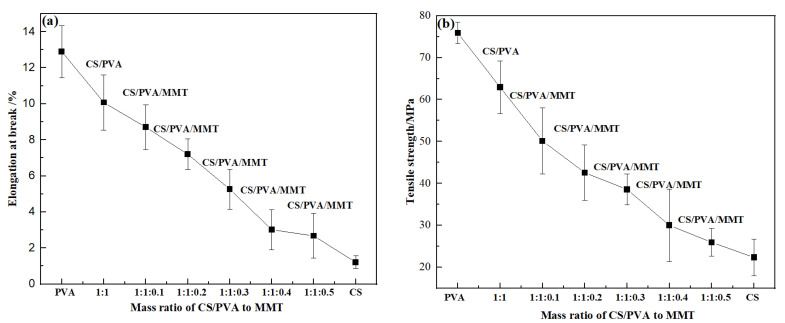
Effect of the mass ratio of CS/PVA to MMT on the mechanical properties of films.

**Figure 2 molecules-27-08984-f002:**
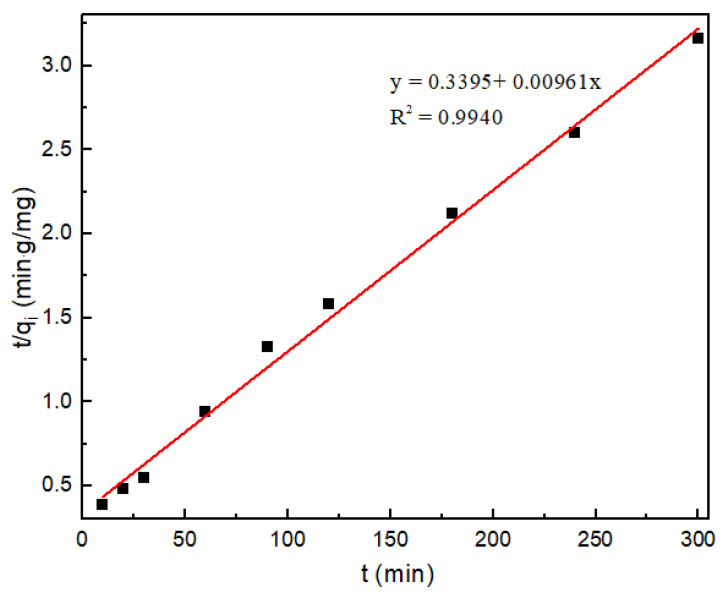
Adsorption of Pd^2+^ by CS/PVA/MMT is consistent with the pseudo-second-order kinetic model.

**Figure 3 molecules-27-08984-f003:**
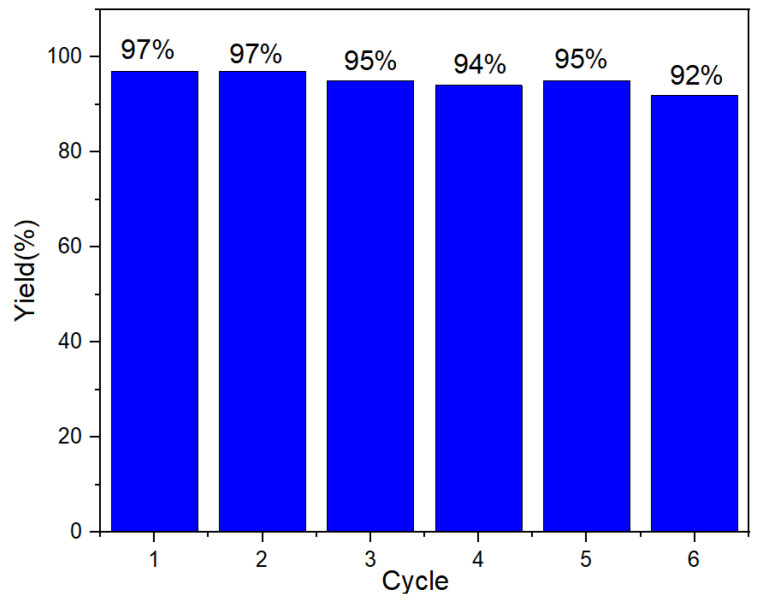
Reuse of CS/PVA/MMT@Pd.

**Figure 4 molecules-27-08984-f004:**
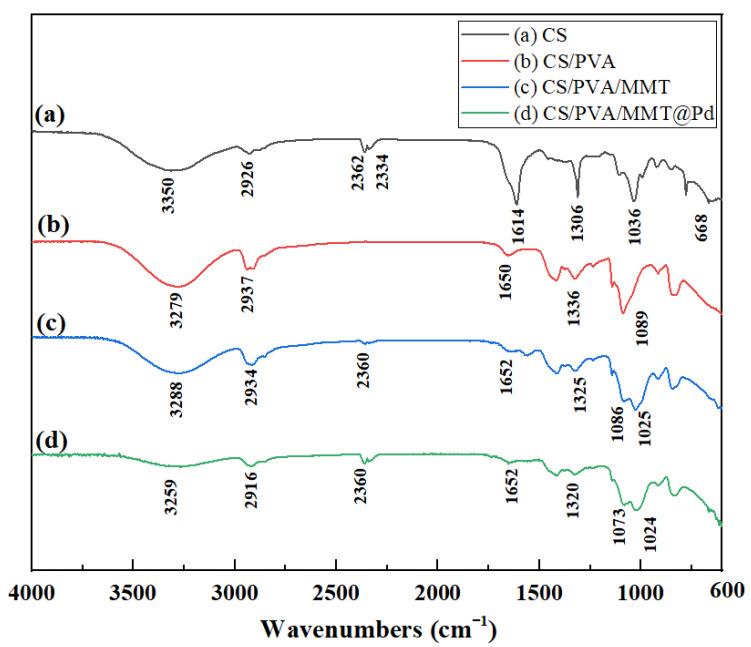
FTIR spectra of different films: (**a**) CS, (**b**) CS/PVA, (**c**) CS/PVA/MMT, (**d**) CS/PVA/MMT @Pd.

**Figure 5 molecules-27-08984-f005:**
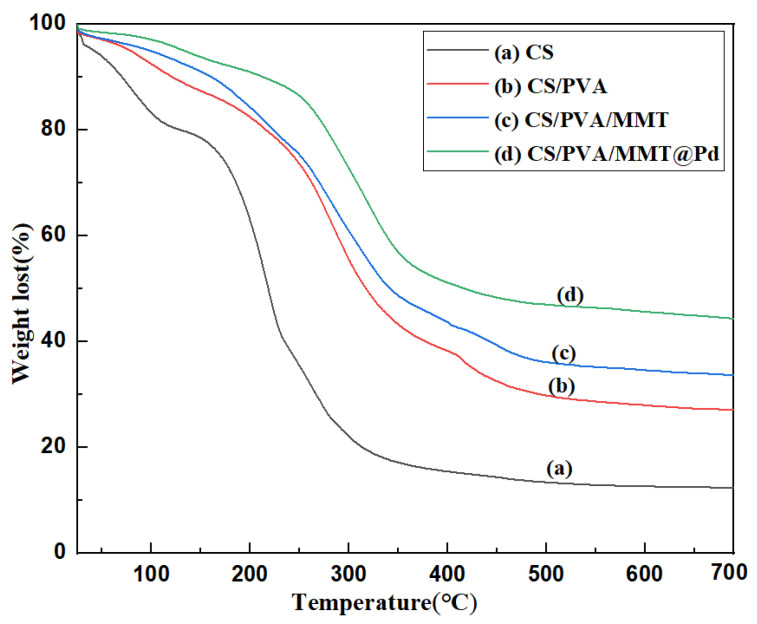
TG curves of different films: (**a**) CS, (**b**) CS/PVA, (**c**) CS/PVA/MMT, (**d**) CS/PVA/MMT@Pd.

**Figure 6 molecules-27-08984-f006:**
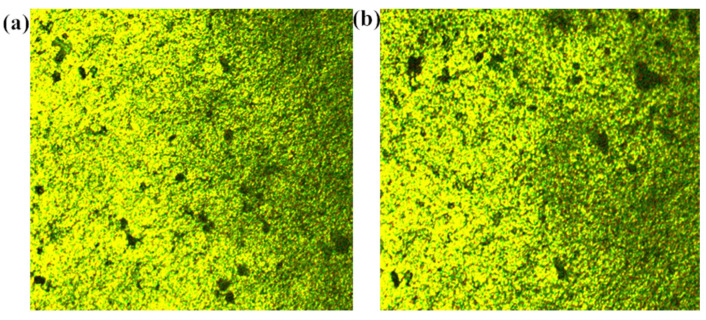
Morphology based on polarizing microscopy (*200): (**a**) CS/PVA/MMT@Pd (no reaction), (**b**) CS/PVA/MMT @Pd (after one reaction).

**Figure 7 molecules-27-08984-f007:**
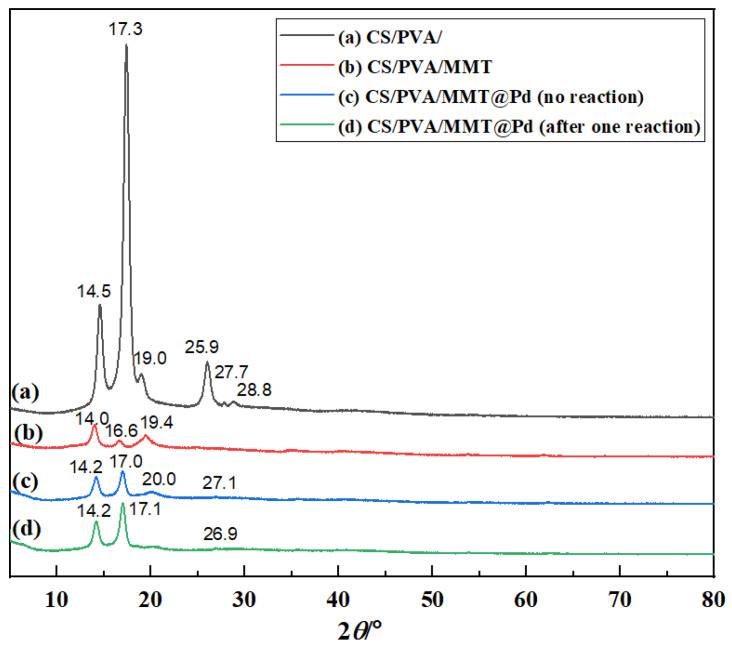
X-ray diffraction patterns of (**a**) CS/PVA, (**b**) CS/PVA/MMT, (**c**) CS/PVA/MMT @Pd (no reaction), and (**d**) CS/PVA/MMT @Pd (after one reaction).

**Figure 8 molecules-27-08984-f008:**
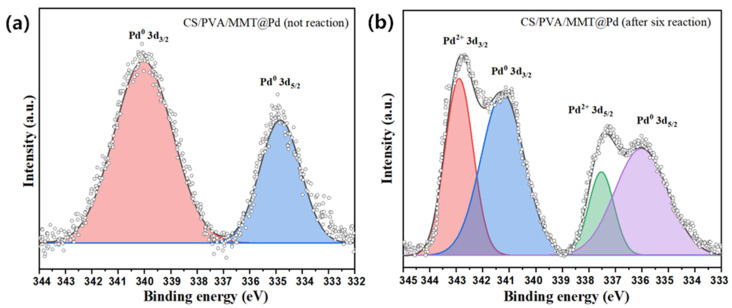
(**a**) XPS analysis of elemental Pd on CS/PVA/MMT@Pd (no reaction) and (**b**) CS/PVA/MMT @Pd (after six reactions).

**Figure 9 molecules-27-08984-f009:**
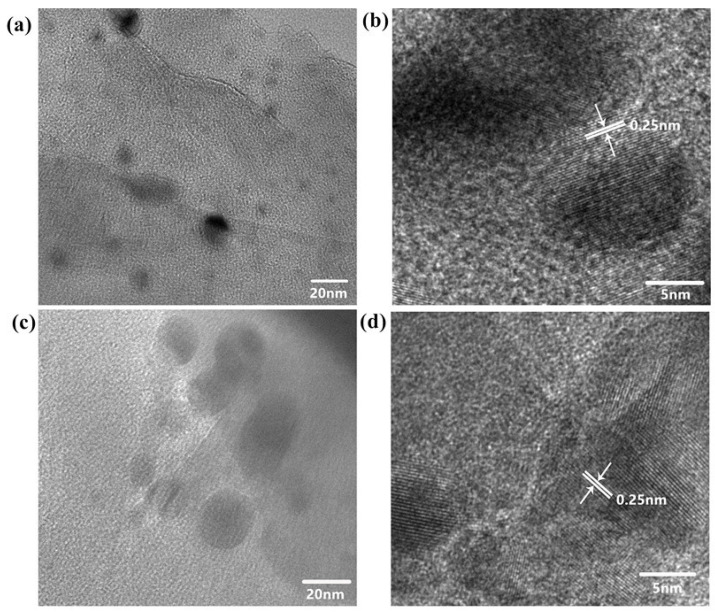
TEM analysis of CS/PVA/MMT@Pd film: (**a**,**b**) CS/PVA@Pd film (no reaction); (**c**,**d**) CS/PVA/MMT @Pd (after six reactions).

**Figure 10 molecules-27-08984-f010:**
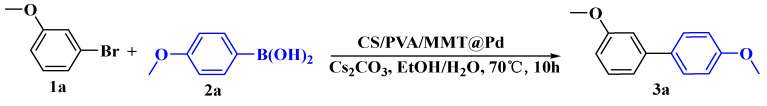
Suzuki coupling reaction of 4-methylbromobenzene with 4-tert-butylbenzeneboronic acid.

**Table 1 molecules-27-08984-t001:** Adsorption capacity and adsorption rate of modified chitosan microspheres for Pd^2+^.

Entry	pH	Original Concentration (*c*_0_, mg/L)	Adsorption Time (min)	Residual Concentration (mg/L)	Absorption Percentage (%)	Absorption Capacity (*q*_i_, mg/g)
1	3	100.00	300	5.09	94.91	94.91
2	2	100.00	300	37.82	62.18	62.18
3	1	100.00	300	60.09	39.91	39.91
4	3	20.15	300	0.99	95.09	19.16
5	3	40.76	300	1.22	97.01	39.54
6	3	60.02	300	1.61	97.31	58.41
7	3	80.07	300	3.73	95.34	76.34
8	3	100.00	10	74.25	25.75	25.75
9	3	100.00	20	58.3	41.70	41.7
10	3	100.00	30	45.22	54.78	54.78
11	3	100.00	60	36.02	63.98	63.98
12	3	100.00	90	32.16	67.84	67.84
13	3	100.00	120	24.11	75.89	75.89
14	3	100.00	180	15.02	84.98	84.98
15	3	100.00	240	7.86	92.14	92.14
16	3	100.00	360	5.11	94.89	94.89
17 ^a^	3	100.00	300	28.7	71.3	71.3

^a^ CS/PVA film.

**Table 2 molecules-27-08984-t002:** Optimization of reaction conditions.

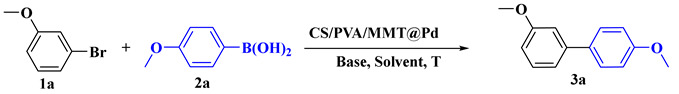
Entry	CS/PVA@Pd(mg)	Solvent (2 mL)	Base (1.2eq)	Time(h)	T/°C	Yield(%)
1	10	MeOH	Na_2_CO_3_	12	60	59
2	10	EtOH	Na_2_CO_3_	12	60	55
3	10	i-PrOH	Na_2_CO_3_	12	60	50
4	10	Toluene	Na_2_CO_3_	12	60	42
5	10	Acetone	Na_2_CO_3_	12	60	10
6	10	CH_3_CN	Na_2_CO_3_	12	60	22
7	10	CHCl_3_	Na_2_CO_3_	12	60	13
8	10	H_2_O	Na_2_CO_3_	12	60	-
9	10	MeOH: H_2_O(1:1)	Na_2_CO_3_	12	60	86
10	10	EtOH: H_2_O(1:1)	Na_2_CO_3_	12	60	88
11	10	i-PrOH: H_2_O(1:1)	Na_2_CO_3_	12	60	80
12	10	Toluene:H_2_O(1:1)	Na_2_CO_3_	12	60	34
13	10	Acetone:H_2_O(1:1)	Na_2_CO_3_	12	60	28
14	10	EtOH: H_2_O(2:1)	Na_2_CO_3_	12	60	92
15	10	EtOH: H_2_O(4:1)	Na_2_CO_3_	12	60	91
16	10	EtOH: H_2_O(1:2)	Na_2_CO_3_	12	60	87
17	10	EtOH: H_2_O(2:1)	Li_2_CO_3_	12	60	75
18	10	EtOH: H_2_O(2:1)	K_2_CO_3_	12	60	93
19	10	EtOH: H_2_O(2:1)	Cs_2_CO_3_	12	60	95
20	10	EtOH: H_2_O(2:1)	-	12	60	-
21	10	EtOH: H_2_O(2:1)	Cs_2_CO_3_ (1.5eq)	12	60	95
22	10	EtOH: H_2_O(2:1)	Cs_2_CO_3_ (2eq)	12	60	94
23	10	EtOH: H_2_O(2:1)	Cs_2_CO_3_	12	70	97
24	10	EtOH: H_2_O(2:1)	Cs_2_CO_3_	12	80	96
25	10	EtOH: H_2_O(2:1)	Cs_2_CO_3_	12	90	92
26	6	EtOH: H_2_O(2:1)	Cs_2_CO_3_	12	70	89
27	8	EtOH: H_2_O(2:1)	Cs_2_CO_3_	12	70	96
28	12	EtOH: H_2_O(2:1)	Cs_2_CO_3_	12	70	97
29	10	EtOH: H_2_O(2:1)	Cs_2_CO_3_	6	70	75
30	10	EtOH: H_2_O(2:1)	Cs_2_CO_3_	8	70	95
31	10	EtOH: H_2_O(2:1)	Cs_2_CO_3_	10	70	97
32	10	EtOH: H_2_O(2:1)	Cs_2_CO_3_	16	70	95

**Table 3 molecules-27-08984-t003:** Substrate expansion of Suzuki reaction.

** 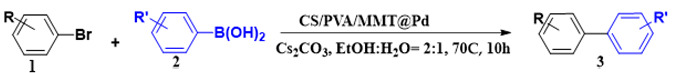 **
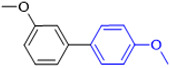	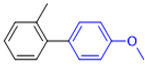	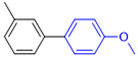	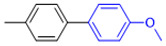
**3a**, 97%	**3b**, 90%	**3c**, 95%	**3d**, 93%
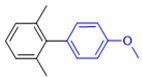	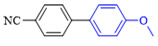	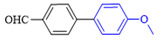	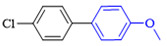
**3e**, 83%	**3f**, 87%	**3g**, 86%	**3h**, 92%
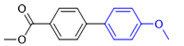	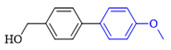	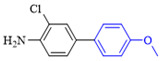	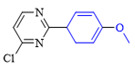
**3i**, 68%	**3j**, 85%	**3k**, 88%	**3l**, 78%
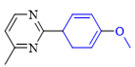	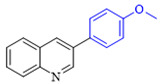	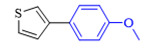	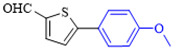
**3m**,52%	**3n**, 86%	**3o**, 92%	**3p**, 90%
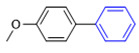	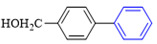	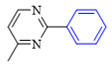	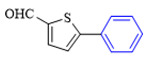
**3q**, 92%	**3r**, 81	**3s**, 77%	**3t**, 85%
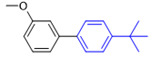	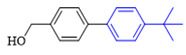	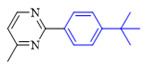	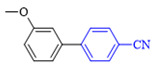
**3u**, 96%	**3v**, 86%	**3w**, 67%	**3x**, 83%
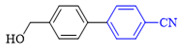	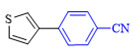
**3y**, 75%	**3z**, 84%

**Table 4 molecules-27-08984-t004:** Chlorobenzene substrate expansion of Suzuki reaction.

** 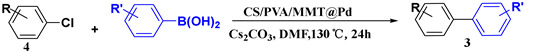 **
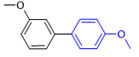	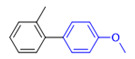	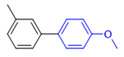	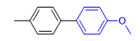
**3a**, 28%	**3b**, 19%	**3c**, 22%	**3d**, 23%
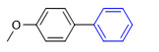	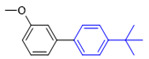	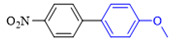	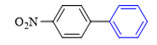
**3q**, 25%	**3u**, 30%	**3aa**, 41%	**3ab**, 37%

**Table 5 molecules-27-08984-t005:** Comparison of catalytic performance of CS/PVA/MMT@Pd catalysts with other catalyst materials.

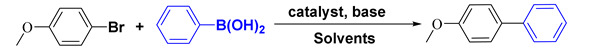
Entry	Catalyst	Solvent	Additive	Temp. (°C)	Time (h)	Yield (%)	Ref.
1	Pt-APA@Fe_3_O_4_/GO	DMF	-	reflux	1	74	[48]
2	Cu(II) PNP pincer complexes	CH_3_CN	-	r.t.	16	69	[49]
3	PdCl_2_	Toluene	-	100	8	99	[50]
4	Pd NPs	Euphorbia granulate leaf extract	-	r.t.	5	92	[56]
5	Pd/C	H_2_O/EtOH	-	r.t.	24	96	[51]
6	PICB-NHC@Pd	EtOH	-	80	36	89	[52]
7	Polyimide@Pd	H_2_O	TBAB	60	1	95	[53]
8	CS@Pd	-	MW	r.t.	0.1	99	[54]
9	CS/starch @Pd	toluene	-	50	0.1	98	[55]
10	CS/PVA/MMT @Pd	H_2_O/EtOH	-	80	12	97	This work

## Data Availability

Supporting data can be obtained from the corresponding authors.

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
