# Peer review of "Preparation of a Montmorillonite-Modified Chitosan Film-Loaded Palladium Heterogeneous Catalyst and its Application in the Preparation of Biphenyl Compounds"

_molecules, 2022, doi:10.3390/molecules27248984_

Round 1
Reviewer 1 Report
The present manuscript by Xiao, Wang and co-workers deals with the preparation and thorough characterization of a hybrid material based on a membrane made with montmorillonite loaded with chitosan functionalized with PVA in which palladium nanoparticles have been immobilized. The above material has been successfully employed as a recyclable heterogeneous catalyst for the Suzuki reaction. In my opinion the investigation has been well conducted and the paper is properly presented and therefore deserves to be published in Molecules. Nevertheless some minor point has to be addressed in order to improve the manuscript.
In the final part of the introduction the sentence: "and the catalyst was easily separated and reused more than 6 times" should be corrected in... reused at least for 6 times.
In Figure 1 the caption on the bottom of the x-axis should be magnified and all the mixtures and materials should be alligned to the corresponding point in the graphic.
The IR discussion should be confirmed after the four IR spectra are normalized, since it is not possible a fine comparison with such difference in transmittance (or absorbance, it is not written).
The actual Pd loading in the fresh catalyst is not reported as well as the Pd loading in all the Suzuki reactions.
In the TGA part the gas used for the analyses should be reported.
There is a problem with figure 7 b, since in the main text is reported that the Pd(0) peaks are centered at 340 and 335 eV but none of the peaks are actually centered at those Binding energies.
Please check and reword the sentence at line 215.
In lines 220 and 221 proton and nonproton solvents should be protic and aprotic, respectively.
In Section 2.5 among the several reactant is present 2,4-dichloropyrimidine: in the title reaction only aryl bromides are supposed to be tested and, by the way, the resulting product is not present in Table 3.
Figure 8 in the X axis should report number of cycle, if in the first run fresh catalyst has been used.
In line 276 reacted catalyst should be used catalyst.
Author Response
Response to Reviewer 1 Comments:
1) In the final part of the introduction the sentence: "and the catalyst was easily separated and reused more than 6 times" should be corrected in... reused at least for 6 times..
Answer: Thank you for your suggestion. We have revised them.
“The results showed that Pd was highly dispersed in the membrane material, with good substrate tolerance and catalytic activity, and the catalyst was easily separated and reused at least 6 times.”
2) In Figure 1 the caption on the bottom of the x-axis should be magnified and all the mixtures and materials should be alligned to the corresponding point in the graphic.
Answer: We regret that such an error occurred and we have corrected it.
3) The IR discussion should be confirmed after the four IR spectra are normalized, since it is not possible a fine comparison with such difference in transmittance (or absorbance, it is not written).
Answer: Thank you for your suggestion and we have corrected it.
4) The actual Pd loading in the fresh catalyst is not reported as well as the Pd loading in all the Suzuki reactions.
Answer: Thank you for your suggestion. We have added the relevant description in section 3.3.
The membrane with sufficient absorption of PdCl2 was selected as the catalytic material and the Pd2+ in the remaining solution after adsorption was tested by ICP-OES and the Pd content in the catalyst was 94.89 mg/g. The actual content of Pd per gram of CS/PVA/MMT film was 0.89 mmol, and the catalyst added for the reaction was 3 mol%. We analyzed the Pd2+ concentration in the solution after the reaction by ICP-OES, but no Pd 2+ concentration was detected.
5) In the TGA part the gas used for the analyses should be reported.
Answer: Thank you for your suggestion. We have revised it: “2.3.2 Thermogravimetric analyzes”.
The thermal stability of the membranes and catalysts was analysed with test conditions set at a heating rate of 10°C/min at N2 flow (20 mL/min).
6) There is a problem with figure 7 b, since in the main text is reported that the Pd(0) peaks are centered at 340 and 335 eV but none of the peaks are actually centered at those Binding energies.
Answer: We regret that such an error occurred and we have corrected it.
After six reaction cycles of CS/PVA/MMT@Pd (Fig. 8b), in addition to the still present characteristic peaks of Pd0 with electron binding energies of 340.98 eV (3d3/2) and 335.98 eV (3d5/2), respectively. characteristic peaks of Pd2+ also appeared with electron binding energies f of 342.58 eV (3d3/2) and 337.08 eV (3d5/2), indicating that the oxygen present in the reaction oxidizes the pd0.
7) Please check and reword the sentence at line 215.
Answer: Thank you for your advice. We have reword the sentence:
“The effect of solvent type, base type, reaction temperature, and reaction time on the reaction were investigated using CS/PVA/MMT@Pd as the catalyst and 3-methoxybromobenzene (1a) and 4-methoxybenzeneboronic acid (2a) as the template substrates.”
8) In lines 220 and 221 proton and nonproton solvents should be protic and aprotic, respectively.
Answer: Thank you for your advice. We have revised it
9) In Section 2.5 among the several reactant is present 2,4-dichloropyrimidine: in the title reaction only aryl bromides are supposed to be tested and, by the way, the resulting product is not present in Table 3.
Answer: Thank you for your advice. We have deleted the 2,4-dichloropyrimidine.
“Among the protic solvents, the higher yields of 3a were MeOH (59%), EtOH (55%), and i-PrOH (50%). In contrast, among the aprotic solvents, the yields were lower than 22% for other solvents such as MeCN, CH3COCH3, and CHCl3, except for toluene as a solvent with a higher yield (42%).”
10) Figure 8 in the X axis should report number of cycle, if in the first run fresh catalyst has been used.
Answer: Thank you for your advice. X axis has been changed to Cycle.
11)- In line 276 reacted catalyst should be used catalyst.
Answer: Thank you for your advice. We have revised it.
“The used catalysts were washed three times with water and ethanol alternately under ultrasonic conditions and then directly entered the cycling experiment.”

Reviewer 2 Report
The ms reports on the preparation, characterization and catalytic properties of novel material comprising palladium supported on chitosan modified with polyvinyl alcohol and montmorillonite. The resulting CS/PVA/MMT@Pd catalyst exhibited excellent catalytic properties in Suzuki reactions of a range of aryl bromides.The work is quite well presented and supported by multiple experiments. I believe that the manuscript is suitable to be published in catalysts after minor revision. Please, see the comments below.
1. It remained unclear what percentage of Pd loading was chosen for the resulting catalyst, and no elemental analysis data (wt% of Pd) to characterize the final CS/PVA/MMT@Pd material was given. Did the authors test catalytic activity of materials with different percentages of palladium?
2. It would be more convenient if description of the structure of recovered catalyst was given after (or inside) the section “2.7. Catalyst reuse” and not together with the characterization of fresh catalyst.
3. Characterization of recovered catalyst should also contain information about active metal (Pd) leaching under turnover conditions. Give, please, data on elemental analysis (wt% of Pd) of fresh and recovered catalyst after catalytic reaction. Information about Pd content in the reaction mixture after separation of the catalyst would be useful as well.
4. Experimental part should contain description of the catalytic reactions procedure.
5. There are also some typos:
-PVA abbreviation was not entered
-‘absorb-ing’ – line 249
-‘Preparation of catalytic materials s’ – line 326
-‘The catalytic material was prepared according to the method of Ref.’ (line 327) – what Ref.?
Author Response
- It remained unclear what percentage of Pd loading was chosen for the resulting catalyst, and no elemental analysis data (wt% of Pd) to characterize the final CS/PVA/MMT@Pd material was given. Did the authors test catalytic activity of materials with different percentages of palladium?
Answer: Thank you for your advice. We have added the relevant description in section 3.3.
“The membrane with sufficient absorption of PdCl2 was selected as the catalytic material and the Pd2+ in the remaining solution after adsorption was tested by ICP-OES and the Pd content in the catalyst was 94.89 mg/g. The actual content of Pd per gram of CS/PVA/MMT film was 0.89 mmol, and the catalyst added for the reaction was 3 mol%. We analyzed the Pd2+ concentration in the solution after the reaction by ICP-OES, but no Pd 2+ concentration was detected.”
- . It would be more convenient if description of the structure of recovered catalyst was given after (or inside) the section “2.7. Catalyst reuse” and not together with the characterization of fresh catalyst.
Answer: Thank you for your advice. We have placed the discussion of catalysts after the catalyst cycle。
“2.6 Catalyst reuse
2.7 Characterization of Catalytic Materials”
- Characterization of recovered catalyst should also contain information about active metal (Pd) leaching under turnover conditions. Give, please, data on elemental analysis (wt% of Pd) of fresh and recovered catalyst after catalytic reaction. Information about Pd content in the reaction mixture after separation of the catalyst would be useful as well.
Answer: Thank you for your advice. We have added the relevant description in section 3.3.
The membrane with sufficient absorption of PdCl2 was selected as the catalytic material and the Pd2+ in the remaining solution after adsorption was tested by ICP-OES and the Pd content in the catalyst was 94.89 mg/g. The actual content of Pd per gram of CS/PVA/MMT film was 0.89 mmol, and the catalyst added for the reaction was 3 mol%. We analyzed the Pd2+ concentration in the solution after the reaction by ICP-OES, but no Pd 2+ concentration was detected.
- Experimental part should contain description of the catalytic reactions procedure.
Answer: Thank you for your advice. We have added it.
“3.6. Suzuki coupling reaction
Fig.9 Suzuki coupling reaction of 4-Methylbromobenzene with 4-tert-Butylbenzeneboronic acid.
3-methoxybromobenzene (1a, 0.3 mmol), 4-methoxyphenylboronic acid (2a, 0.36 mmol), Cs2CO3 (0.36 mmol), and CS/PVA/MMT@Pd (10 mg, 3mol%) were weighed into a reaction tube in air, and 2 mL of mixed solvents were added as the reaction solvent at 70℃. After the reaction had stopped, the extract was split and spun dry to collect the primary product, then a mixture of ethyl acetate and petroleum ether (v:v=1:100-1:8) was used as a drenching agent and the product was collected and separated by chromatography on a column and the final product structure was verified by NMR.”
- There are also some typos:
-PVA abbreviation was not entered
-‘absorb-ing’ – line 249
-‘Preparation of catalytic materials s’ – line 326
-‘The catalytic material was prepared according to the method of Ref.’ (line 327) – what Ref.?
Answer: We regret that such an error occurred and we have corrected it.
(1) In line 56 of the four paragraph of section 1, we have added the PVA abbreviation:“The addition of polyvinyl alcohol (PVA) to the catalytic carrier material can effectively improve the mechanical properties of the material by taking advantage of its excellent film-forming properties.”
(2) The catalytic material was prepared according to the method of Ref [57].
57.Zheng, K.; Yang, F.; Huang, Z.; Zhan, Y.; Xiao, Z.; Li, W.; Wang, W.; Qin, C., Preparation of chitosan film-loaded palladium catalyst materials and their application in Suzuki coupling reactions. Journal of Materials Research and Technology 2022, 20, 3905-3917.
